# Short text classification with machine learning in the social sciences: The case of climate change on Twitter

**Karina Shyrokykh**  *, **Max Girnyk, Lisa Dellmuth**

Department of Economic History and International Relations, Stockholm University, Stockholm, Sweden

* karina.shyrokykh@ekohist.su.se

## Abstract

To analyse large numbers of texts, social science researchers are increasingly confronting the challenge of text classification. When manual labeling is not possible and researchers have to find automatized ways to classify texts, computer science provides a useful toolbox of machine-learning methods whose performance remains understudied in the social sciences. In this article, we compare the performance of the most widely used text classifiers by applying them to a typical research scenario in social science research: a relatively small labeled dataset with infrequent occurrence of categories of interest, which is a part of a large unlabeled dataset. As an example case, we look at Twitter communication regarding climate change, a topic of increasing scholarly interest in interdisciplinary social science research. Using a novel dataset including 5,750 tweets from various international organizations regarding the highly ambiguous concept of climate change, we evaluate the performance of methods in automatically classifying tweets based on whether they are about climate change or not. In this context, we highlight two main findings. First, supervised machine-learning methods perform better than state-of-the-art lexicons, in particular as class balance increases. Second, traditional machine-learning methods, such as logistic regression and random forest, perform similarly to sophisticated deep-learning methods, whilst requiring much less training time and computational resources. The results have important implications for the analysis of short texts in social science research.

## Introduction

Contemporary society is characterized by the ever-greater production of textual data. Citizens post on social media, politicians give speeches which are commented upon live, and international organizations publish reports. Such activities provide social scientists with large amounts of freely available data, enabling them to advance theories in areas such as international relations, comparative politics, media studies, and public opinion.

An example is the field of social media studies which is growing apace, as social networks have become an important aspect of people's lives. The popularity of social media platforms means that they often serve as a means of communication for citizens living in distant places, civil society organizations, business representatives, and political institutions. Such societal and political actors tend to make use of social media platforms to exchange information with

**Data Availability Statement:** The data underlying the results presented in the study are available here: https://zenodo.org/record/7633599#.Y-IbTS8w1qs.

**Funding:** This work was supported by the project: "Glocalizing Climate Governance: The role of Integrated Governance for a Just and Legitimate Adaptation to Climate Risks (GlocalClim)" funded by Formas under grant number 2018-01705 (Svenska Forskningsrådet Formas, Grant/Award Number: 2018-01705). The funder had no role in study design, data collection and analysis, decision to publish, or preparation of the manuscript.

**Competing interests:** The authors have declared that no competing interests exist.

the broader public. Social media has thus been defined as a social tool shaping new and unmanaged communication dynamics [1]. For example, the microblogging site Twitter is often used to package information in short messages [2–4].

In the study of short texts, researchers are often interested in categorizing or labelling texts according to their topic, relevance, or stance before carrying out further analysis. This text classification process is a key challenge for two main reasons. First, manual classification of large numbers of texts might not be feasible and automated methods might be required. However, little is known about the relative performance of such methods in the context of social science research [5]. Second, concepts in the social sciences are often ambiguous and complex. This makes it difficult for automated classifiers to categorize texts based on those.

A case in point is the topic of climate change and the risks it causes for societies and ecosystems. Over the past thirty years, framings of climate change as a policy challenge have proliferated and differentiated across sectors and scales. Political debates about "climate adaptation", "climate mitigation", "carbon emissions", "climate-induced displacement", "disaster risk reduction", and "climate vulnerability" all illustrate the complex nature of the concept [6, 7]. When classifying texts about climate change and its governance, researchers need to know how the concept of climate change is being referred to in the specific context in which it is being studied.

Several approaches to automated text classification are known. Lexicon-based methods are the most popular among social scientists [8–10]. These are based on lexicons or dictionaries of meaningful words related to a topic of interest, crafted by experts, and made available to the research community. Because such dictionaries are not always available, machine-learning (ML) algorithms are being increasingly applied. The two main categories of ML methods are supervised and unsupervised learning.

In unsupervised ML methods, the computer may cluster the words in texts around unknown categories. These clusters, and the categories they might indicate, are then interpreted by the researcher [11]. However, such approaches might not be a good choice when seeking to classify texts using labels that have been defined *a priori*. When the categories are known, supervised learning is more appropriate [12, 13]. Here, researchers first manually label a smaller dataset, which is then used to train a model that identifies the specified categories in a larger dataset in an automated way. Remarkably, in recent years, a subset of these methods, deep learning (DL), has achieved human-level performance carrying out certain tasks, such as image recognition [14] and text translation [15]. DL methods have proven very efficient at handling complex non-linear relationships within data, but require large amounts of advanced computations [16].

The choice of algorithm used for ML depends principally on the task at hand, but should also be motivated by model performance. In this article, we compare several alternatives to find the best-performing algorithm. Drawing on the literature on text classification in the social sciences [5, 9, 17–19], we identify the most widely-used lexicon-based and supervised ML text-classification methods. We then investigate the performance of these methods in a typical social science research setting: a small labeled dataset with rare-event data. Our empirical focus is on Twitter communication about climate change, which is studied in a rapidly growing body of interdisciplinary social science research [2, 20–24].

To delineate the empirical material, we make use of the Twitter communication of eight international organizations in different policy areas that are known to be central in communicating about climate change, and which are comparable in their communication as they are all part of the United Nations (UN): the Food and Agriculture Organization (FAO), the Office for the Coordination of Humanitarian Affairs (UNOCHA), the UN Development Programme (UNDP), the UN Office for Disaster Risk Reduction (UNDRR), the UN Environmental

Program (UNEP), the UN International Children's Emergency Fund (UNICEF), the UN High Commissioner for Refugees (UNHCR), and the World Health Organization (WHO) [25, 26]. Not only the communication by these organizations on climate change, but also political debates, in general, are often complex and relate to multiple policy areas at the same time. The case of climate change debates on Twitter is not unique in this regard. This means that the findings reported in this article about the use of text classification methods for this case can also be applicable to other political debates.

In this article, we make three main contributions to social science research using short texts. First, we identify relevant dictionary-based and ML methods, showing how these methods might be employed for text classification. Second, we show, in the context of small labeled datasets exhibiting class imbalance, that supervised ML methods perform better than lexicons, in particular as class balance increases. Third, we find that traditional ML methods, such as logistic regression and random forest, perform similarly to sophisticated DL methods, whilst requiring much less training time and computational resources. In all, these contributions have important implications for the use of text classifiers in the social sciences, as we elaborate in the conclusions.

The remainder of this article proceeds as follows. We begin by discussing the text classification problem that we address with automated text classification methods. We then review the most commonly used automated text classification methods. Next, we present the dataset of UN agencies' tweets and compare the performance of these methods in analysing the texts within this dataset. Finally, we summarize our findings and sketch avenues for future methodological research on text classification.

## Automated text classification methods

In this section, we review existing automated text-classification methods. These methods rely on a similar workflow consisting of the collection of data and various text preprocessing techniques, which we discuss in detail in the S1 Appendix. We start with the lexicon-based classifier. After this, we move on to the basics of the supervised-learning approach to classification. This is followed by an overview of relevant traditional ML algorithms. Finally, we discuss advanced neural classifiers.

To describe the methods, we use the following notation. Boldface letters, such as $a$, denote vectors. Capital boldface letters, such as $A$, denote matrices and calligraphic letters, such as $\mathcal{A}$, denote sets. Estimates are denoted as $\hat{a}$. A vector transpose is denoted as $a^T$. Lower indices, such as in $a_j^{(i)}$, are used to denote features, whilst upper indices denote observations. The probability distribution of a variable $a$ is denoted as $p(a)$, and $p(a|b)$ designates the conditional probability distribution of $a$ given $b$ has been observed, where we deliberately do not distinguish between the random variables and their realizations.

### Lexicon-based classifier

Lexicon-based classification is the most widely used approach to text classification in the social sciences. It is an unsupervised technique that is based on the filtering or labeling of texts with the help of a lexicon, or a list of relevant key terms (e.g., [8, 10]). Typically, the lexicon that is used is created by an expert in the area of interest, who has knowledge of the terminology used in context in this area.

When we use lexicon-based classification to categorize Twitter data from UN agencies, the content of tweets is tokenized (or split up into separate words) and then inspected. If one or more of the tokenized words in the tweet matches with key terms contained within the lexicon, then the tweet is classified as dealing with climate change. If no words match, then the tweet is

classified as not dealing with climate change. The number of tokenized words which need to match for positive classification is a design parameter.

There are many possible ways to classify tweets with respect to the topic of climate change. However, there is no standard lexicon for this type of classification. For this article, in order to benchmark the ML methods with a reasonable classifier, we select the lexicon from [3]. It consists of the following key terms: "climate", "climatechange", "globalwarming", "climaterealists" and "agw"(an abbreviation for "anthropogenic global warming"). The lexicon has a decent performance and constitutes a good baseline for our empirical performance assessment.

## Supervised approach to text classification

An efficient alternative to lexicon-based classifiers is the use of supervised ML methods. This class of methods requires a manually labeled dataset $\mathcal{D}$ (see Eq (1) and Fig 3 in the S1 Appendix). By fitting a set of $N$ examples given in this labeled dataset, the methods obtain a model that approximates the predictor function. To fit the model, we choose a mapping $\hat{f}(\cdot)$ that minimizes the empirical *average loss*. This means that, in fitting the model, the computer attempts to minimize the difference between the predicted value and the actual observation. This can be formally expressed as:

$$L(\hat{f}) = \frac{1}{N}\sum_{i=1}^{N}\ell\left(\hat{f}(\boldsymbol{x}^{(i)}), y^{(i)}\right),\tag{1}$$

where $\ell(\hat{f}(\boldsymbol{x}^{(i)}), y^{(i)})$ is some chosen loss function that reflects the closeness of the model prediction $\hat{f}(\boldsymbol{x}^{(i)})$ for features $\boldsymbol{x}^{(i)}$ to the true label $y^{(i)}$ for observation $i$.

The model $\hat{f}(\cdot)$, being a function of the features $\boldsymbol{x}$, is also a function of a set of its own parameters $\boldsymbol{w} = [w_0, \ldots, w_{K-1}]$. The fitting of the model is therefore carried out by searching the parameter vector $\boldsymbol{w}$ that minimizes the average loss in Eq (1). This process is referred to as the training of the model. By monitoring loss $L(\hat{f})$ we can track the in-sample performance of the model, or more specifically how closely $\hat{f}_{\boldsymbol{w}}(\cdot)$ corresponds with the data to which it is being fitted. Because the model learns the "average" mapping from features into labels, we also expect the model to have a decent out-of-sample performance (i.e., performance on new and as-yet-unseen examples). Unfortunately, the in-sample performance of a model might not be indicative of its out-of-sample performance, as we shall see below.

If the model is too simple (i.e., it has too few parameters $\boldsymbol{w}$ to capture the behavior of the data), it is unable to fit the data well and has poor in-sample performance. This is known as underfitting and is an indicator that the model is poor in general. To avoid underfitting, more complex models have to be considered. The true predictor $f(\cdot)$ can be a complicated relation between features and labels. To approximate it well, the model $\hat{f}_{\boldsymbol{w}}(\cdot)$ may sometimes need to be complicated and include many parameters $K$. For example, a linear model of this kind could be given by the following expression: $\hat{f}_{\boldsymbol{w}}(\boldsymbol{x}) = w_0 + w_1\varphi_1(\boldsymbol{x}) + \cdots + w_{K-1}\varphi_{K-1}(\boldsymbol{x})$, where $\varphi_k(\boldsymbol{x})$ is some function of features $\boldsymbol{x}$ for $k = 1, \ldots, K - 1$. More complicated models (e.g., polynomials) might involve even larger numbers of parameters.

The more complex the model $\hat{f}_{\boldsymbol{w}}(\cdot)$, the greater the possibility to decrease the empirical loss $L(\hat{f})$ in Eq (1) and improve its in-sample performance on the given dataset $\mathcal{D}$. However, when the number of parameters becomes too high (relative to the number of examples $N$ in the dataset), the training process might start confusing the noise in the data with the behavior of the feature variables. The model $\hat{f}_{\boldsymbol{w}}(\cdot)$ becomes tied to particular examples in the dataset and does not generalize to unseen data. That is, the in-sample performance of the model on the dataset

becomes highly misleading and does not reflect its out-of-sample performance, which is known as overfitting. Thus, a general rule for picking the model is the following: the model should be complex enough to overcome underfitting, but simple enough to not reach overfitting.

To detect overfitting, the entire dataset $\mathcal{D}$ can be split into two parts: a training set $\mathcal{D}_{\text{train}}$ and a validation set $\mathcal{D}_{\text{valid}}$. This makes it possible to validate the out-of-sample performance of a model on the validation set. Then several models can be fitted to the training set, while also validating their out-of-sample performance using the validation set. This validation can be used as a basis for selecting a model that is complex enough to generalize to new examples and performs the best in terms of average loss among all other considered models. Following this, the best model can be retrained using the entire dataset (minus the test set), whilst also ensuring that it generalizes well when applied to as-yet-unseen data.

This approach makes it possible to validate the models and select the best one in terms of the out-of-sample loss. However, it might still not be possible to assess the actual performance of the model on as yet unseen data. This is because the process of model selection contaminates the dataset, injecting a dependency between the model and the data based on which it was selected. Hence, the validation loss is not representative of the actual performance on yet unseen data. To tackle this problem, it is necessary to make yet another partition of the dataset $\mathcal{D}$. That is, prior to cutting subsets $\mathcal{D}_{\text{train}}$ and $\mathcal{D}_{\text{valid}}$ from the dataset $\mathcal{D}$, another subset of data, called a test set $\mathcal{D}_{\text{test}}$ is withheld from the validation set. This test set is not touched during the training and validation, and is used purely for assessing the classifier's performance on unseen data.

## Traditional machine-learning classifiers

By traditional ML classifiers we mean a collection of ML algorithms which have previously been developed for classification tasks. Those are built on different principles and have different computational complexities. Their performance depends on the task at hand. In this section, we will review the most popular ML classifiers and specify the variants of the classifiers that will be used for our subsequent performance comparisons. In our analysis, we use the Sci-Kit-Learn implementations [27] of the traditional ML algorithms, whereas for the deep neural classifiers, we use the TensorFlow implementations set up via the Keras API [28]. For each of the methods, we test the three available vectorizers, as well as optimize the number of features to maximize the classifier's performance.

**Logistic regression.**   Logistic regression (LR) is a classification algorithm that was invented in [29] and has been widely applied in text classification [30, 31]. The classifier models a binary random variable representing the label $y$ given the data $\boldsymbol{x}$. The model, parameterized by weights $\boldsymbol{w}$, is given by

$$\hat{f}_{\boldsymbol{w}}(\boldsymbol{x}) = \sigma\left(w_0 + \sum_{j=1}^{M} w_j x_j\right), \tag{2}$$

where $\sigma(\cdot)$ is the *sigmoid function* representing the conditional probability of a positive class label given the observed features $x_1, \ldots, x_M$.

An example of a LR classifier in action is shown in Fig 1 on exemplary synthetic data. The two classes are separated by a decision boundary, which is determined by the equal-probability line on the sigmoid-based probability map (see the black solid line on Fig 1). For all data points with probability above the decision threshold we predict $\hat{y} = 1$, while for the rest we predict $\hat{y} = 0$.

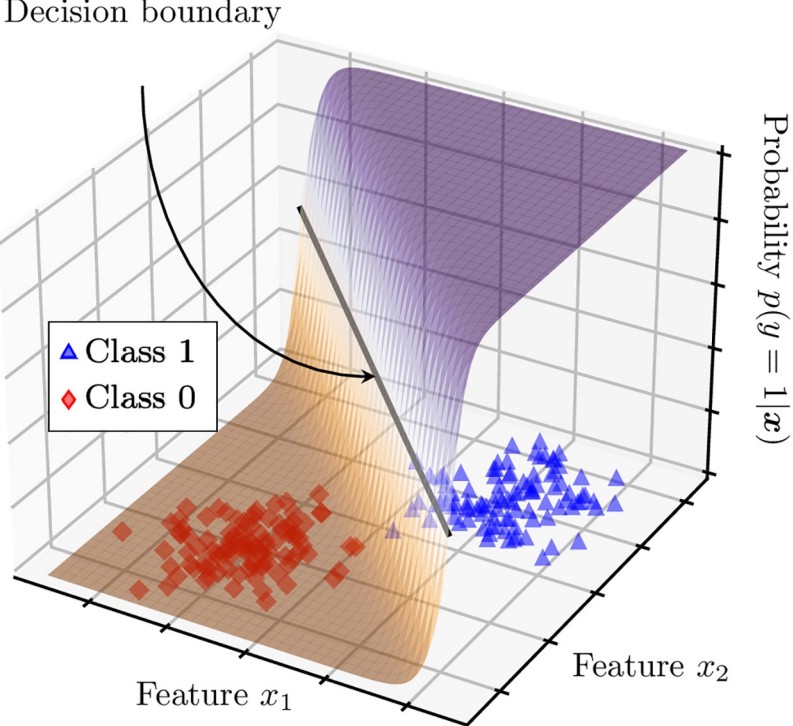

**Fig 1. Illustration of a logistic regression classifier.** The sigmoid-based surfaces shows the probability of the positive class given the observed data.

The advantages of the LR classifier are its simplicity, efficient training, interpretability and the absence of assumptions about the class distribution. The downsides are overfitting for a large number of features (compared to the number of observations) and sensitivity to multicollinearity among predictors.

A classifier is characterized by a set of *hyperparameters* which refer to parameters related to the model's architecture and whose values are set before the learning process begins. This is in contrast to the model's own *parameters* which are learned during the training. The SciKit-Learn implementation of the LR classifier has several hyperparameters, including regularization strength and maximum number of iterations. For our numerical performance assessment, these are tuned to get the best classification performance.

**Support vector machines.** Support vector machines (SVMs) were first proposed in [32] for linearly separable problems and later generalized to non-linear problems in [33]. SVMs have been frequently used for the classification of political texts [12, 13, 18]. They are sometimes seen to be *the* classification method to be used for short texts in social sciences [8]. Unlike the LR method which provides a probability as its output, the output of an SVM gives the direct prediction of the label.

SVM classification works by fitting a hyperplane that separates the classes and maximizes the separation margin [32]. *Support vectors* refer to those samples that are the closest to the separating hyperplane. These are the only samples that impact the SVM training, as they are most likely to cause misclassification. The margin is given by the length of the projections of the supporting vectors onto the vector of parameters $w$ which is perpendicular to the separating hyperplane. The average loss is given by Eq (1) with a hinge loss function.

The above intuition only works for cases where classes are linearly separable (see Fig 2a). In the general case, when classes may overlap in such a way that it is not possible to fit a separating hyperplane between their data points, the SVM principle can be extended to *soft margins*. This allows some observations to reside on the other side of the hyperplane [34]. Furthermore, a non-linear transformation $\phi(\cdot)$ can also be applied to the data, expanding the feature space to a higher dimension in which there may be a linearly separating hyperplane. This transformation can be done, e.g., by adding higher-order polynomial terms. However, a problem with this approach is that it might explode the computational complexity of the algorithm and lead to overfitting. In order to overcome this problem, the "kernel trick" can be invoked. Namely, by means of Lagrangian duality, the maximization of the soft margin is reformulated in terms of scalar products between data points. There are *kernel functions* $k(\cdot)$ where $k(\boldsymbol{x}^{(i)}, \boldsymbol{x}^{(j)}) = \phi(\boldsymbol{x}^{(i)})\phi(\boldsymbol{x}^{(j)})$—e.g., polynomial, Bessel, radial basis functions (RBFs). This means that it is possible to skip computing the data transformation $\phi(\boldsymbol{x})$. Instead, only computing the kernel function will give the dot products in the transformed feature space for the optimization. The obtained separating hyperplane translates back to the original feature space as a non-linear decision boundary (see Fig 2b that illustrates an SVM with an RBF kernel on synthetic data).

The advantages of the SVM classifier are its operation efficiency with high-dimensional data, ability to model non-linear decision boundaries, and efficient handling of small datasets. The downsides are its sensitivity to the choice of the kernel function, high computational complexity for very large datasets, and the lack of probabilistic interpretation (unlike in LR).

Because it is known that text classification is usually linearly-separable [12], we use an SVM implementation with a linear kernel for our empirical analysis. The regularization hyperparameter, defining the "softness" of the margin, is tuned to maximize the performance of the SVM.

**Random forest.** Random forest (RF) classifiers [35, 36] are widely used for text classification (see, e.g., [37, 38]). The algorithm uses the so-called *decision tree* approach [39, 40]. The latter relies on the intuition that, instead of seeking a complicated mapping function $\hat{f}(\boldsymbol{x})$, one can partition the feature space into disjoint regions and fit a simpler model in each of those. This is done during training by recursively splitting the space of possible values for each feature at a certain threshold. For example, at a given iteration, the entire input space is split into a pair of half-spaces $\{\boldsymbol{x} : x_j \geq t_j\}$ and $\{\boldsymbol{x} : x_j < t_j\}$, where $t_j$ is a threshold for a certain feature $j$. The split is done in a greedy way to minimize some loss function (e.g., misclassification error, Gini impurity, or entropy), without looking ahead at future splits. This process is repeated on each of the half-spaces recurrently until a stopping criterion is fulfilled (e.g., there is no information gain from further splits). In this way, a binary "decision tree" is constructed, which sets the thresholds for the classification of new samples.

Once the training is done, a new observation $(\boldsymbol{x}, y)$ is passed through the entire decision tree. As it passes through the tree, the observation experiences a sequence of learned conditional statements (greater or lower than a threshold) for each feature $x_j$ where $j = 1, \ldots, M$. At the bottom of the tree (in a leaf node, where there is no further split), the prediction is obtained as the majority class of the data points satisfying all the conditions along the path. Illustrations of predictions with decision trees are shown as one block in Fig 3. Each circle depicts a conditional statement over a feature with respect to a threshold. The thick lines depict the paths of observations as they travel through the decision tree, satisfying all the conditions on their way. The numbers below the trees indicate the predicted classes.

An issue with decision trees is that they typically overfit training data. A widely-used way to improve generalizability is to average the predictions of a number of decision trees by means of bagging [41]. With bagging, several bootstrap training sets $\mathcal{D}_1, \mathcal{D}_2, \ldots, \mathcal{D}_B$ are

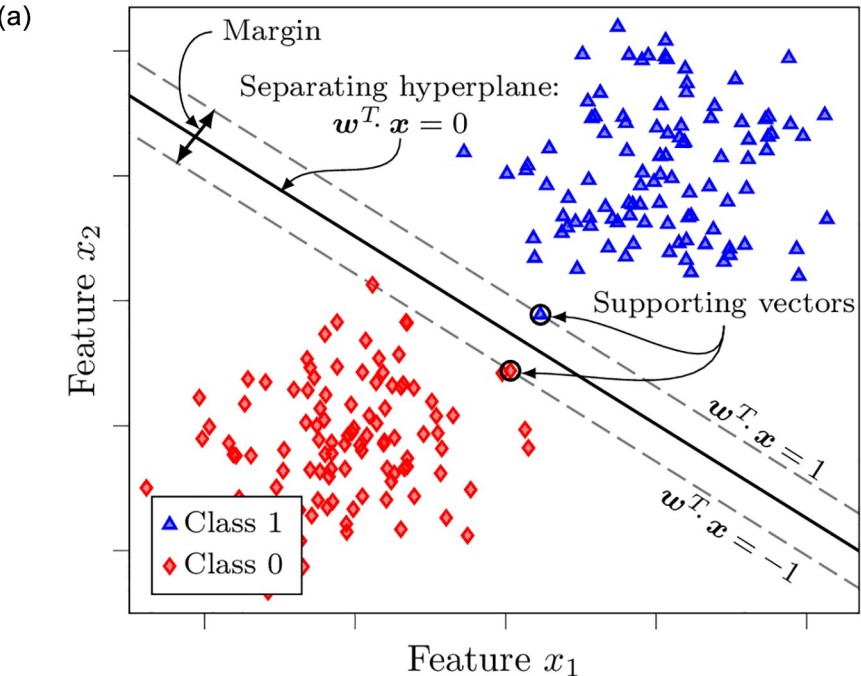

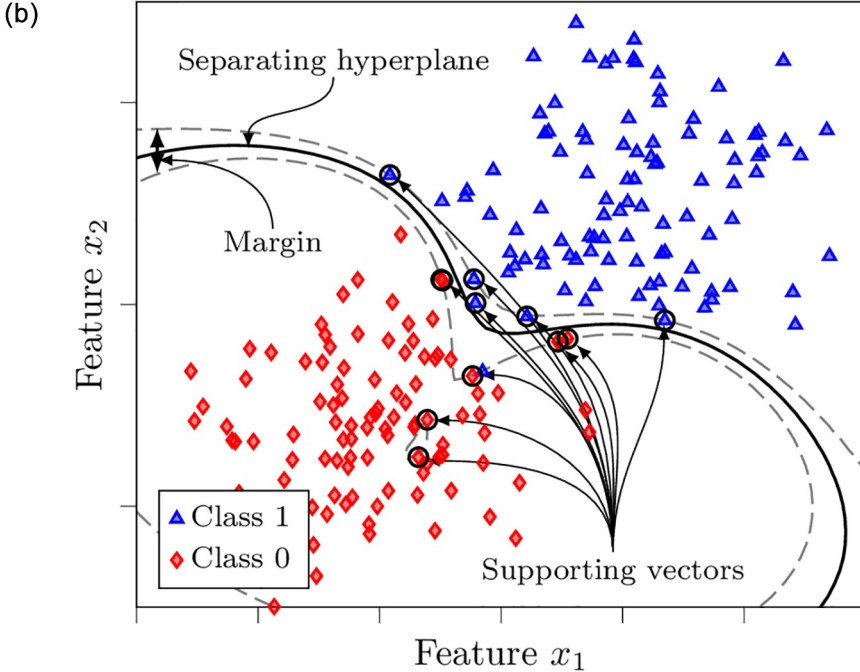

**Fig 2. Illustration of a support vector machine classifier.** A hyperplane is fitted to separate the two classes and maximize the margin. Linear kernel (a) is used for the case of linearly separable classes, while radial basis function kernel (b) is used for the case with class overlap.

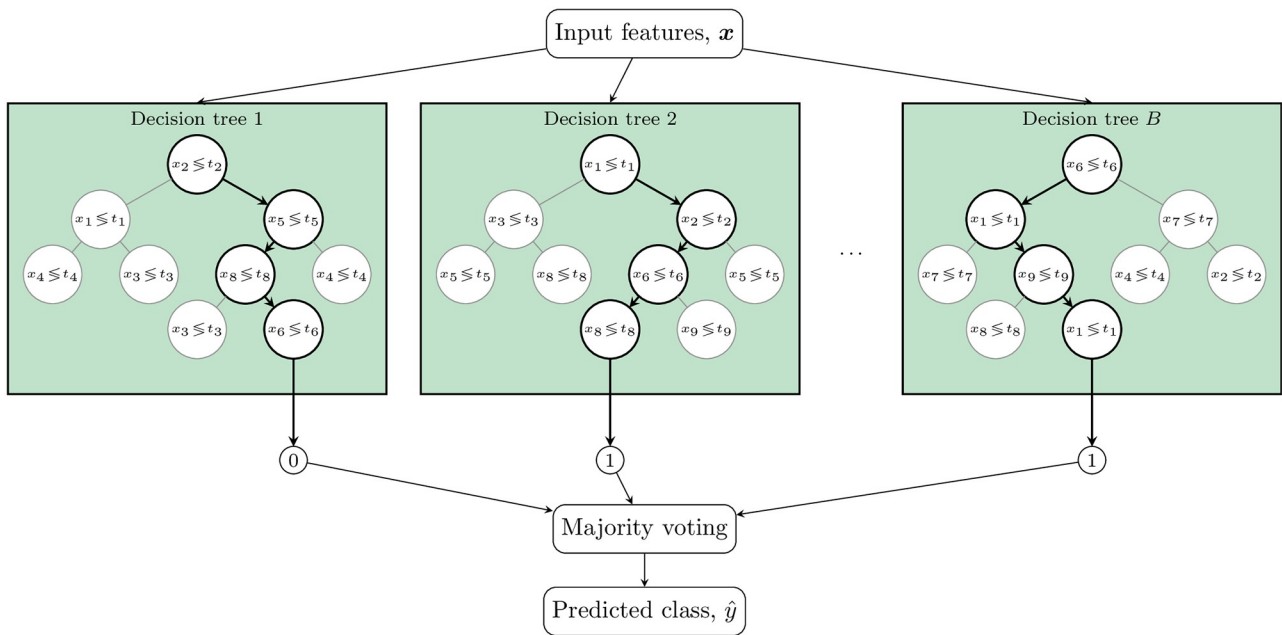

**Fig 3. Illustration of a random forest classifier.** The classifier consists of a collection of decision trees, each making its own prediction regarding the class of an unseen data point. The predictions are then used in a majority voting for producing the final prediction.

randomly sampled (with replacements). A decision-tree model is then fitted to each of these training sets. The predictions from the models are combined, e.g., using majority voting over decision trees in the forest. This improves the model's generalizability by reducing the variance of the classifier, without increasing its bias. The RF approach is an extension of bagging that also randomly selects subsets of features used in each data sample. Fig 3 illustrates an RF classifier in action. A total of $B$ decision trees are formed by considering various subsets of available features. Each decision tree makes its own prediction on its own subset of the training data. Then majority vote defines the final prediction label of the random forest classifier.

The RF classifier has the advantages of good interpretability, possibility to assess the importance of input features, and efficient avoidance of overfitting with many trees. The main disadvantage of this method is its computational complexity and slow speed with a large number of trees, which limits its applicability for real-time operation.

For our empirical analysis, we adjusted the maximum depth of the decision trees and the number of estimators in order to optimize the RF classifier's performance.

**k nearest neighbors.** The $k$ nearest neighbors (KNN) algorithm [42] is a simple classifier which has been widely used by researchers for text categorization (see, e.g., [43, 44]). In contrast to other ML algorithms, KNN does not have a training phase. Instead the training set is simply stored and the actual computation is deferred to the prediction phase. In a nutshell, given an unlabeled observation $\boldsymbol{x}$, the algorithm calculates the distances $d(\boldsymbol{x}, \boldsymbol{x}^{(i)})$ to training observations $\boldsymbol{x}^{(i)}$, for $i = 1, \ldots, N$, in the feature space and finds $k$ nearest neighbors to $\boldsymbol{x}$. The classes of these neighbors are used to weigh the available classes for the given observation. That is, majority voting is performed on the observations within the set of $k$ nearest neighbors. Fig 4 illustrates a KNN classifier in action on synthetic data.

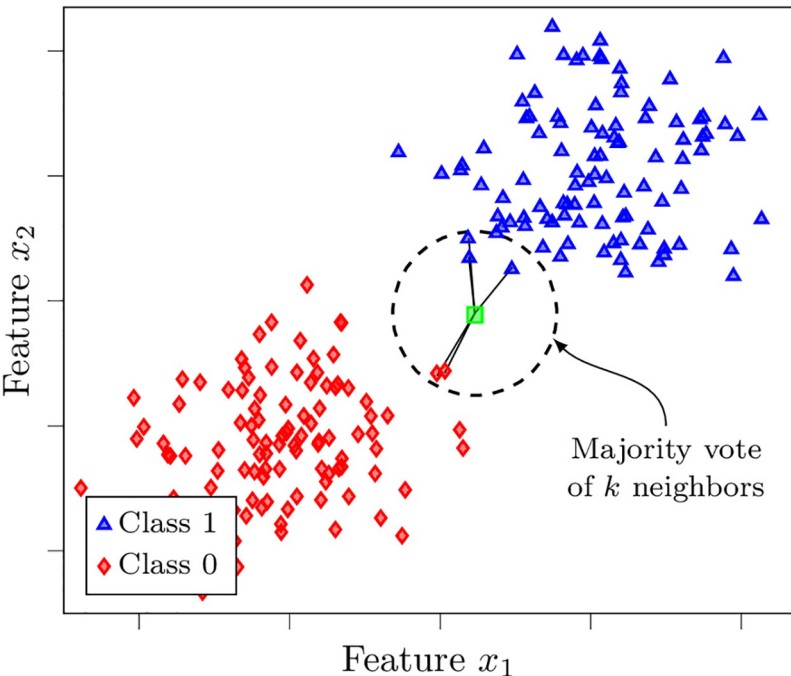

**Fig 4. Illustration of a *k* nearest neighbors classifier.** For an unseen data point, the distances towards its *k* nearest neighbors are computed, and the majority class gets assigned.

In our analysis, we use KNN with uniform weights. Because of the majority voting procedure, we set the hyperparameter *k* to an odd number. We sweep *k* and another hyperparameter, leaf size, to optimize the KNN performance.

**Naïve Bayes.** Naïve Bayes (NB) is a probabilistic classifier whose basic idea is to use the joint probabilities of words and categories to estimate the conditional category probabilities given a data point [45]. It has been successfully used for text classification [46, 47]. It is called "naïve" because it makes an assumption that features are conditionally independent given the label. This assumption simplifies the computations carried out by the NB classifier, when compared to a non-naïve Bayes classifier, because the NB classifier does not use word combinations as predictors.

Given an observation $\boldsymbol{x}$, such as a short text, the NB classifier assigns the most probable category $\hat{y}$ according to

$$\hat{y} = \text{argmax}_{c \in C} \{p(c|\boldsymbol{x})\} = \text{argmax}_{c \in C} \{p(\boldsymbol{x}|c)p(c)\}, \tag{3}$$

due to Bayes' theorem. The term $p(\boldsymbol{x})$ is not dependent on class $c$ and hence skipped. Since it is difficult to estimate $p(\boldsymbol{x}|c)$, the naïve Bayes assumption is made. Namely, we factorize this conditional distribution as $p(\boldsymbol{x}|c) = p(x_1|c)p(x_2|c) \cdots p(x_M|c)$, which simplifies the computations. Moreover, $p(c)$ is estimated as the fraction of the tweets in class $y$ in the training set. Meanwhile, $p(x_j|c)$ is estimated as the relative count of feature $x_j$ in class $c$ to all features, with applied Laplace smoothing.

The advantages of the NB classifier are its simplicity, good scalability, quick training and insensitivity to irrelevant data. The disadvantages of the method are the assumption of independent predictor features and zero-probability problem for features present in the test set, but not occurring in the training set.

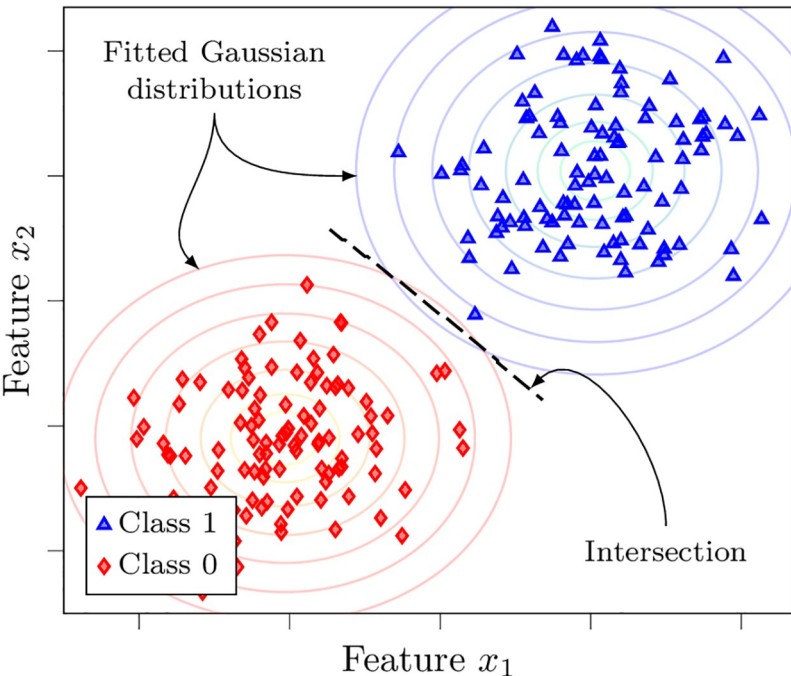

**Fig 5. Illustration of a Gaussian naïve Bayes classifier.** The classifier fits two Gaussian distributions (depicted with contour plots) to the categorized labels in the training set. The decision boundary is determined by the point where the probability densities for the two categories take the same value.

There are several versions of the NB classifier (e.g., Gaussian, Bernoulli, multinomial). The operation of a Gaussian NB classifier on exemplary data is demonstrated in Fig 5. Studies have reported that a multinomial mixture model shows improved performance scores on several data collections, when compared to other commonly used versions of the NB approach [46]. Because of this, for our empirical analysis, we use a multinomial NB classifier with the smoothing hyperparameter tuned to achieve the best performance.

## Deep learning

*Deep learning* (DL) is a prominent supervised ML framework for data-driven predictive modeling. Despite its promising classification capacities, DL continues to be a much less used approach to text classification in social science, when compared to its use in such fields as computer science. A DL classifier can be thought of as a black box: data go in, decisions come out (see Fig 6a). Inside the black box there are a large number of parameters that are learned during the training. In this way, the DL model approximates a predictor function that describes the observed data. Because of the way its operations resemble those of the human brain, the broader class of methods to which DL belongs is often referred to as *artificial intelligence* (AI).

A central element of DL is the concept of an artificial *neural network* (NN) which describes the inner-workings of the DL black box. It consists of a layered structure of units conducting mathematical operations on their inputs and passing along the results (see Fig 6b). The usefulness of an NN lies in its ability to infer a function from observations. In this way, the DL model can approximate the predictor function that describes the observed data which is essential for both classification and regression tasks. The idea of an NN was first proposed by Hebb [48] as an abstraction of a real biological network of neurons in the mammalian brain.

(a)

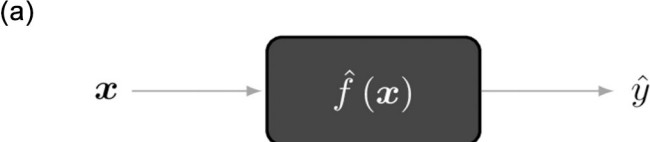

(b)

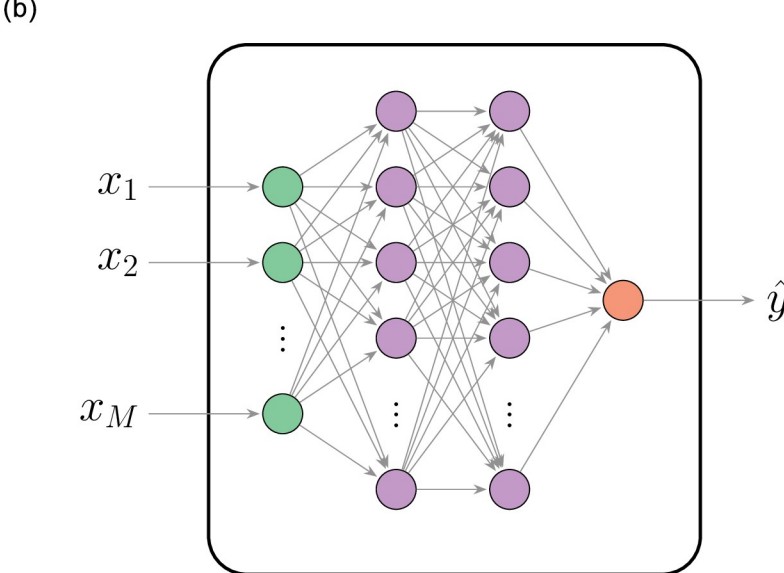

**Fig 6. Illustration of a deep-learning model.** The classifier provides a complex non-linear function (a) that maps an input feature vector into a predicted label. The function is learned from the training set by adjusting the parameters of a set of internal units arranged in, e.g., a layered fully-connected structure (b). (**a**) Deep-learning model as a black box. (**b**) Fully-connected deep neural network.

A biological NN comprises numerous layers, each consisting of a set of neurons (see Fig 4 in the S1 Appendix). Each neuron consists of *dendrites* that receive input signal pulses, a *cell body* that performs a transformation of the incoming combination of pulses, and an *axon*. The axon carries the output pulse through its trunk (covered with isolating *myelin sheaths*) to its ramified terminal, which has a set of *synapses* that connect to the dendrites of the upstream neurons in the network. The functioning of biological NNs has been studied and formalized independently in [49, 50].

Each neuron in the network conducts a non-linear operation on the superposition of the signals received by its dendrites. These signals are weighted by the strength of the connection of its synapses of the upstream neurons. The process carried out by each neuron can therefore be modeled as a simple unit computing the weighted sum of its inputs $x_1, \ldots, x_M$ with weights $w_{j,1}, \ldots, w_{j,M}$, adding a bias term $b_j$ and applying a non-linear transformation $a(\cdot)$, called an *activation function* on the weighted sum (see Fig 4a in the S1 Appendix). This unit, first proposed in [51], is referred to as an *artificial neuron* and constitutes the basic building block of every NN.

Information travels through the NN from the input layer towards the output layer, passing through all neurons on its way. For binary classification tasks, the output layer consists of a single neuron that outputs the probability of the observation belonging to one of the classes.

The layers in between are called *hidden layers*, as they are not directly visible from the outside of the black box. An NN is referred to as a *deep* NN if it contains two hidden layers or more. Note that NNs could equally well be adapted for regression and multi-class classification tasks. For the latter, the last layer would consist of the same number of neurons as there are classes, each having a softmax activation function and outputting the probability of belonging to that class.

It has been proven that sufficiently deep NNs are able to approximate any arbitrarily complex function $f(\cdot)$ [52]. Hence, deep NNs are particularly useful in applications where there are large amounts of data and significant computational resources available, and where manual analysis is tedious or the performance of traditional ML algorithms is unsatisfactory. In the context of text classification, NNs were first applied in [53, 54]. Only recently have they started to be used for political science research [55, 56]. There are various neural architectures available for this task. We review the most relevant ones below.

**Fully-connected neural networks.** The first attempt to combine artificial neurons into a layer was made by Rosenblatt [57]. He devised the concept of a *perceptron*, a binary classifier mapping an input feature vector into one of several available classes. Formally, given a feature vector $x$, the perceptron outputs

$$\hat{y} = a\left(\sum_{j=1}^{M} w_j x_j\right), \tag{4}$$

where $\{w_j\}_{j=1}^{M}$ is the set of weights, and $a(\cdot)$ is the activation function. This step is referred to as the *forward pass*. Given an actual training output $y$, the algorithm readjusts its weights according to the *gradient descent* rule

$$w_j \leftarrow w_j + \alpha(y - \hat{y})x_j, \tag{5}$$

where $\alpha$ is the so-called learning rate which determines the convergence speed of loss minimization. This adjustment process is referred to as the *backward pass*. The process continues iteratively until the loss function in Eq (1) is below an acceptance threshold. If the classification problem is linearly separable, the perceptron is guaranteed to converge, and the predictor function is given by the forward pass with the learned weights $w_j$ for all $j = 1, \ldots, M$.

In a nutshell, the perceptron is an NN with a single layer. To improve its performance, Werbos [58] proposed stacking several perceptrons in consecutive layers. That study coined the concept of a multi-layer perceptron, also known as a *fully-connected* NN (FCNN). FCNNs exhibit a much better learning ability than that of the perceptron, and are the simplest form of a general class of feedforward NNs, which are neural architectures without cycles, being the most common architectures used nowadays.

An FCNN comprises a set of fully-connected (or dense) layers stacked in a line topology (see Fig 6b). Due to the presence of multiple layers, an FCNN is capable of having multiple levels of abstraction. An example is its use for facial recognition [56]. Based on an input image, the first layers of the NN capture the simple characteristics of the image, such as oriented edges or corners. Then further layers react to more complicated shapes, such as noses or eyes. Finally, the last layers are able to detect the face itself. In this way, adding extra layers to the NN can enable solutions to a lot of otherwise non-separable problems.

For a deep FCNN with $P$ hidden layers, the input-output relation at each layer $p = 1, \ldots, P + 1$ is given in a matrix form as

$$z_p = a_p(W_p z_{p-1} + b_p), \tag{6}$$

where $z_{p-1}$ and $z_p$ are vectors of the inputs and outputs of layer $p$, respectively. Then $W_p$ is the matrix of weights between the input and output neurons of the layer, while $b_p$ is the vector of bias terms of the given layer. Moreover, $a_p(\cdot)$ is the activation function, which is often chosen as sigmoid, hyperbolic tangent (tanh), or rectified linear unit (ReLU). Also note that here the input of the first layer is given by the input feature vector $z_0 = x$. Meanwhile, during the training, the output of the last layer in the network is set to the true label, i.e., $z_{P+1} = y$. For a new data point, the output of the last layer provides the predicted label, i.e., $z_{P+1} = \hat{y}$.

Learning for feedforward NNs is usually done by means of backpropagation. The NN parameters (i.e., weights and biases) for each layer are adjusted based on the adopted loss function which captures the difference (or error) between the observation, $y$, and the output of the forward pass based on current parameters $\hat{y}$. The adjustment is done based on the loss function (1). This learning process is often carried out by means of the stochastic gradient descent, or any other known optimization algorithms used for learning, such as, e.g., Adam [59] and RMSProp [60]. This involves recursively propagating the gradients of the parameters backwards, from the last to the first layer, similarly to Eq (5). It is noteworthy that a mini-batch version of the gradient update is often used for more robust convergence. More concretely, the training dataset is split into small batches for which the model loss is calculated and model parameters are updated. In this way, the batch updates are computationally more efficient because they do not need the training data to be held in the computer's memory.

The advantages of FCNNs are their ability to solve complex non-linear problems, efficiency of handling large amounts of data, and quick predictions after the slow process of training is completed. Their disadvantages are slow training, poor interpretability due to their black-box nature, large numbers of parameters due to the fully-connected structure, and ignoring spatial information by accepting only vectorized inputs.

For our analysis, we have used the same architecture for all NNs under consideration, with difference in only a single layer that is particular to the given NN. For FCNN, this layer comprised a fully-connected layer with a certain number of units and a ReLU activation function. The number of neurons was optimized, alongside other hyperparameters, such as batch size, number of epochs, dropout rate, learning rate, regularization strength, embedding dimension, and maximum vocabulary length.

**Convolutional neural networks.** Convolutional neural networks (CNNs) refer to another subclass of feedforward NNs that are designed to capture spatial and temporal dependencies through the application of kernels. CNNs have the benefit of having a reduced number of parameters and reusable weights. The use of CNNs has become a state-of-the-art approach for the analysis of images [16]. They have also been shown to be useful for text classification [55, 61].

A CNN consists of two parts, where the first part is dedicated to feature extraction, while the second part performs the actual classification by means of an FCNN (see Fig 5 in the S1 Appendix). CNNs often start with an *embedding layer* that maps a discrete variable to a vector of continuous numbers. This is done by means of a single fully-connected layer (or a set thereof) that is either learned during the training or substituted by a pre-trained embedding. In the subsequent *convolution layer*, a set of fixed-size kernels slide through the list of embeddings, performing a convolution operation. This layer functions as the CNN's feature extractor because it learns to find local spatial features in the output of the embedding layer. The size of the kernel is the number of embeddings it sees at once multiplied by the length of an entire word embedding. The output of this layer is a set of feature maps that serve as the input for the subsequent *pooling layer*. Zero-padding is often used in this step, surrounding the input so that a feature map does not shrink.

The pooling layer then subsamples the feature maps input to it. It does this by selecting a single element from each region of the feature maps covered by its filter. The pooling layer operates on each feature map independently. This reduces the size of the feature representation, thereby effectively reducing the amount of parameters in the network, whilst preserving the most prominent features. Oftentimes, a *dropout layer* is also applied as a means to improve generalizability. By destroying the learned co-dependencies between neurons that compensate for errors from the previous layers, overfitting is prevented at a cost of longer training. The output is then fed into a fully-connected layer (or an FCNN) that conducts the classification based on the features extracted by the previous layers.

CNNs have important advantages, such as automatic feature extraction, with the possibility of using those for new tasks, and weight sharing, which leads to a reduced number of parameters. They also have lower computational complexity than FCNNs. Their drawbacks are ignoring the position and orientation of the object in their predictions, their need for large amounts of data, and long training times.

In our performance comparisons, the CNN architecture includes an embedding layer, a dropout layer, a convolution layer and a max-pooling layer with output flattened to match a single neuron for final classification. We optimize the same hyperparameters as we tuned for the FCNN, except for the number of units. Instead, for a convolution layer we tune the number of filters and the size of the kernel.

**Long short-term memory neural networks.**   An alternative to feedforward NNs for text classification are *recurrent neural networks* (RNNs). These are NN architectures that are comprised of a chain of repeating modules. This allows previous outputs to be used as the next inputs of the same neural layer. Because of this feature, RNNs are able to incorporate context from previous steps when dealing with a current step. They can use their internal state—also known as memory—to process sequences of inputs with variable lengths.

Unfortunately, plain versions of RNNs suffer from the vanishing gradient problem during training with backpropagation. This limits the number of passes information can make through RNNs, making them impractical. A solution to this problem, so-called *long short-term memory* (LSTM) NNs, has been proposed [62]. This is nowadays considered as one of the breakthroughs in the field of DL.

An LSTM neural network is an RNN with an LSTM cell in place of a neural layer (see Fig 6 in the S1 Appendix). This can let information pass through the cells, as well as adding/removing information while it is passing through a cell. This is regulated by various gates, each comprising a neural layer and an element-wise multiplication operation. The neural layers are equipped with a sigmoid activation function, which determines the amount of information that should pass through (between 0% and 100%). The forget gate then controls what information is kept and what is thrown away from the cell state. It comprises a neural layer with a sigmoid activation. Next, the input gate decides what information to update and what to store. This is followed by a layer with hyperbolic tangent (tanh) activation that outputs a vector of new candidate values that could be added to the cell state to update it. After this, the output gate decides what information to output, which is followed by yet another tanh gate that takes the current state as an input and multiplies the result with the output of the output gate.

The benefits of LSTN networks are their ability to handle long-term dependencies, avoidance of the vanishing-gradient problem and efficient handling of complex sequential data. The disadvantages are their need for large amounts of data and, consequently, slow training and high computational complexity, as well as their worse performance on data that is not of sequence type or contains a lot of noise.

In practice, a bi-directional model [63] is added on top of the LSTM layer. This runs separately in forward and backward directions, concatenating the predictions of both models, and

sending them to an LR model. This allows one, at any point in time, to preserve information from both the past and the future. For our experiments, we optiimze the same parameters as for the FCNN and CNN. There is a slight difference for the bi-directional LSTM layer, where we tune the number of units and the layer dropout rate.

## Results

Next, we evaluate how well the methods discussed in the previous section perform in the task of tweet classification. For the sake of reproducibility, notebooks containing the codes are available at our repository [64].

### Data

For the analysis, we have collected a dataset [65] with tweets from eight Twitter accounts: @UNOCHA, @UNDP, @FAO, @WHO, @UNICEF, @UNDP, @UNDRR, and @Refugees. As elaborated in the introduction, these accounts correspond to international organizations in the UN system that deal with climate change in different policy areas [25, 26].

The tweets were downloaded and parsed via the Twitter Academic Research API. In total, the dataset contains 222,191 tweets posted by the eight UN organizations from their official accounts. This number represents the total number of tweets by these eight selected UN organizations between the beginning of their tweeting history and the end of 2019. The data comprises all tweets posted by the official accounts of the selected UN organizations, whether or not these tweets are concerned with climate change. Each tweet is considered as an observation in our dataset.

From the entire dataset, 5,750 tweets were randomly selected and labeled manually as either "climate change-related" or "not climate change-related". Three research assistants, with training in climate governance, each labeled 2,000 tweets, working independently from each other. After this manual coding, we obtained a dataset containing 5,750 observations, see Table 1. We "lost" 250 observations due to overlaps between the three datasets produced by the research assistants, with 125 overlaps between each pair. This overlap between the datasets was used to track the agreement between the three research assistants' coding.

The inter-coder reliability was high: on average, the coders agreed on 96% of coding decisions, which corresponds to Cohen's $\kappa = 0.95$ [66], viz., very high inter-coder agreement [67]. Two senior researchers trained the research assistants and supervised the coding. In this manual coding, tweets about climate change mitigation and adaptation were both classified as climate change-related, i.e., assigned label "1".

**Table 1. Summary of the collected dataset.**

| Organization | Account | Start date | End date | Tweets |
|---|---|---|---|---|
| FAO | @FAO | Jan. 2009 | Dec. 2019 | 753 |
| UNDP | @UNDP | Jul. 2009 | Dec. 2019 | 1,199 |
| UNDRR | @UNDRR | Oct. 2010 | Dec. 2019 | 256 |
| UNEP | @UNEP | May 2009 | Dec. 2019 | 540 |
| UNHCR | @Refugees | Jun. 2008 | Dec. 2019 | 1,114 |
| UNICEF | @UNICEF | Jul. 2009 | Nov. 2019 | 910 |
| UNOCHA | @UNOCHA | Jul. 2011 | Jul. 2019 | 366 |
| WHO | @WHO | May 2008 | Dec. 2019 | 612 |
| | | | **Total** | **5,750** |

## Performance assessment

For model training and performance assessment, we split the dataset into a training set (85%) and a test set (15%). To avoid data leakage, we made sure that the test set is never used until the final step of the performance evaluation. The latter was carried out by comparing the labels outputted by a classifier to the ground truth (labels marked by our research assistants).

Additionally, a part of the training set (ca. 15%) was taken as a validation set for hyperparameter tuning.

There are many known optimization algorithms used for hyperparameter tuning, e.g., Bayesian optimization [68], random search [69] and metaheuristics, such as genetic algorithm [70], simulated annealing [71], differential evolution [72] and swarm intelligence [73]. For tuning the hyperparameters of all the methods considered herein, we have chosen to use Bayesian optimization, i.e., learning the probability distribution of the model loss conditioned on a given hyperparameter from the historical data gathered from its previous iterations. In particular, we used the tree-structured Parzen estimator [74] for our evaluations.

For the traditional ML methods, in addition to model-specific parameters, we chose the best feature vectorization approach (count vectorizer vs. TF-IDF). For DL models, the Keras tokenizer is used for the conversion of texts to sequences. We also treated the maximum number of considered features as a hyperparameter and tuned it for all models. All NNs have a common architecture. They contain the following layers: an embedding layer, a dropout layer, a layer with a certain type of units (FCNN, CNN or LSTM) and a ReLU activation function, a flattening layer (where applicable), and a fully connected layer. The actual weights and biases of the neural network are optimized with the help of the Adam optimizer [59]. The weights and biases of the NNs are initialized using the Xavier rule [75] to avoid problems with vanishing gradients.

During the training of NNs, we tracked the loss values on the training and validation sets (binary cross-entropy is used as a loss function). We picked the model that achieves the lowest validation loss. We have also used the technique of early stopping to catch the minimum of the validation loss before potential overfitting kicks in. Also, note that in all DL-based methods we utilize self-learned embeddings. An investigation of the improvements from pre-trained embeddings (see, e.g., [76, 77]) has been left for future research. Further details on the NN architectures and optimized hyperparameters can be found in our repository. Finally, it is worth noting that we run all our experiments in the Google Colab environment [78], using graphic processing units (GPUs) to accelerate the training of the DL classifiers.

## Performance metrics

The choice of performance indicators is central to our analysis. For this purpose, it is necessary to consider that our dataset is imbalanced, as only ca. 9% of the tweets in the dataset are about climate change. Prediction accuracy, defined as the share of correct predictions out of all predictions made, is typically used as a performance metric for classification problems. However, for an imbalanced dataset, accuracy becomes a misleading metric. For instance, for our dataset, a simple guess that a tweet is not about climate change would yield 91% accuracy. However, this is not useful for the classification task at hand. Therefore, when choosing evaluation criteria, one has to take into account the dataset imbalance (see [79] for a discussion).

Thus, we relied on alternative performance indicators based on the so-called confusion matrix [79]. For each tweet in the test set, based on the observed features $x$, the classifier under consideration makes a categorization $\hat{y}$. It classifies the tweet as either "positive" (about climate change, $\hat{y} = 1$) or "negative" (not about climate change, $\hat{y} = 0$). When we compared the predicted label $\hat{y}$ (or a classifier's decision) with the corresponding manually-coded label $y$, each

|        |                      | Predicted | |
|--------|----------------------|-----------|---|
|        |                      | Positive ($\hat{y} = 1$) | Negative ($\hat{y} = 0$) |
| True   | Positive ($y = 1$)   | True positives ($N_{TP}$) | False negatives ($N_{FN}$) |
|        | Negative ($y = 0$)   | False positives ($N_{FP}$) | True negatives ($N_{TN}$) |

**Fig 7. Confusion matrix containing counts of four sets of test observations.** The categorization of each observation is based on the comparison of the predicted label with the true one.

observation falls into one of four sets: true negatives (with $N_{TN}$ entries), true positives (with $N_{TP}$ entries), false positives (with $N_{FP}$ entries), false negatives (with $N_{FN}$ entries)—see Fig 7.

Based on the confusion matrix, precision refers to the fraction of correctly classified examples among all the examples classified as being in a category. Meanwhile, recall refers to the fraction of correctly classified examples amongst all the examples that belong to a category in the dataset. These two metrics can have values between 0 and 1, and are computed as

$$P = \frac{N_{TP}}{N_{TP} + N_{FP}}, \qquad R = \frac{N_{TP}}{N_{TP} + N_{FN}}. \tag{7}$$

The higher the value of each of these two metrics, the better the performance of the classifier. However, each of these metrics are in a trade-off against the other. Changing the classifier's cutoff threshold (which determines which of the categories an observation falls into) can improve precision—at the expense of recall. And vice versa, improvements in recall can result in losses in precision.

Given this trade-off, the preferred metric in this article is the so-called *F1 score* as a compromise metric [79, 80] which is computed as the harmonic mean of the two aforementioned metrics, i.e., $F_1 = 2PR/(P + R)$. Various alternative metrics have been used in the literature for assessing the performance of classifiers on imbalanced datasets, e.g., ROC AUC [81], PR AUC [82] and MCC [83]. However, all of these come with their respective advantages and disadvantages. For the purpose of imbalanced classification, there is no single performance metric that can capture all aspects and tell us exactly how well a particular classifier performs. It therefore might be more informative to separately report several metrics, including precision and recall [84]. Nonetheless, in this article, we focus on the F1 score as the primary performance metric as it has been widely used in the existing literature to assess the performance of classifiers applied to imbalanced datasets.

## Performance results

Table 2 presents the performance of the ML algorithms under consideration in terms of all of the metrics discussed above given the original collected dataset. The results suggest that the best-performing approaches are the LR and RF classifiers. This finding is remarkable since these classifiers beat all the sophisticated neural classifiers at much lower computational complexity.

At the same time, we note that the performance scores are generally quite low (the best method, LR, having $F_1 \approx 65\%$ only). This is likely due to the characteristics of the dataset: imbalance and a rather small size. Also notable is the comparatively high performance of the lexicon-based classifier, which reaches $F_1 \approx 62\%$. Another interesting result is the poor performance of the KNN algorithm (with as little as $F_1 \approx 29\%$). Finally, it is noteworthy that the advanced DL methods (FCNN, CNN and LSTM) require significantly longer training (see Table 3).

**Table 2. Classification performance of considered classifiers on the original collected dataset.**

| Method | Accuracy | Precision | Recall | F1 score | AUC ROC | AUC PR | MCC |
|---|---|---|---|---|---|---|---|
| LR | 0.957077 | 0.829268 | 0.531250 | **0.647619** | 0.913945 | 0.694443 | 0.643570 |
| RF | 0.952436 | 0.716981 | 0.593750 | **0.649573** | 0.924048 | 0.685173 | 0.627497 |
| Lexicon | 0.957077 | 0.909091 | 0.468750 | **0.618557** | 0.732525 | 0.709352 | 0.635333 |
| SVM | 0.941995 | 0.606061 | 0.625000 | **0.615385** | 0.881050 | 0.641829 | 0.584106 |
| LSTM | 0.944316 | 0.642857 | 0.562500 | **0.600000** | 0.863565 | 0.591703 | 0.571686 |
| FCNN | 0.938515 | 0.593220 | 0.546875 | **0.569106** | 0.890488 | 0.615837 | 0.536574 |
| NB | 0.937355 | 0.586207 | 0.531250 | **0.557377** | 0.919447 | 0.617646 | 0.524492 |
| CNN | 0.935035 | 0.568966 | 0.515625 | **0.540984** | 0.891115 | 0.605069 | 0.506828 |
| KNN | 0.938515 | 1.000000 | 0.171875 | **0.293333** | 0.887287 | 0.598238 | 0.401461 |

*Note*: The results are sorted by the preferred metric, highlighted in bold font.

The poor classification performance of all the classifiers discussed above can likely be attributed to class imbalance in the dataset. To investigate the influence of this imbalance, we prepare a balanced dataset. To do this, we supplement our original dataset with additional tweets taken from Kaggle's "Twitter Climate Change Sentiment Dataset" [85]. This dataset contains 43,943 tweets about climate change, manually labeled for sentiment. We are not interested in the sentiment labels, but instead take all of these tweets as ones about climate change (or labeled with "1"). We pick enough samples from the Kaggle dataset to balance out our own dataset and run the same algorithms on the new balanced dataset. We ensure that the same train/test split (i.e., 85%/15%) is maintained and the training and test sets in the original dataset remain parts of the new training and test sets, without mixing between the two. In this way, we make sure that the models see only new data during the performance evaluation stage.

Table 4 shows how the classification algorithms perform on the new, balanced dataset. It is notable how dramatically the overall performance scores have improved when the classification methods are applied to the balanced dataset. Almost all methods now exhibit very high F1 scores, larger than 96%. The only outliers are the KNN algorithm ($F_1 \approx 90\%$) and the lexicon-based classifier ($F_1 \approx 86\%$ only). We note that the poor performance of the KNN classifier, compared to other ML algorithms, is in line with the conclusions of [86]. Meanwhile, the

**Table 3. Training and execution times (in seconds) of considered classifiers on the original collected dataset.**

| Method | Training time [s] | Execution time [s] |
|---|---|---|
| NB | 0.004 | 0.008 |
| KNN | 0.004 | 0.156 |
| Lexicon | 0.246 | 0.102 |
| LR | 0.255 | 0.004 |
| SVM | 0.485 | 0.136 |
| RF | 0.897 | 0.160 |
| CNN | 119 | 0.200 |
| FCNN | 133 | 0.319 |
| LSTM | 185 | 1.050 |

*Note*: Here, training time is the time required to train an ML model on the training set. Meanwhile, execution time is the time required to run a trained ML model on the test set for the given dataset size and split (85%/15%).

**Table 4. Classification performance of considered classifiers on the artificially balanced dataset.**

| Method | Accuracy | Precision | Recall | F1 score | AUC ROC | AUC PR | MCC |
|---|---|---|---|---|---|---|---|
| CNN | 0.970551 | 0.977128 | 0.963659 | **0.970347** | 0.990883 | 0.992459 | 0.941192 |
| RF | 0.970551 | 0.992136 | 0.948622 | **0.969891** | 0.990770 | 0.993153 | 0.942009 |
| LR | 0.969298 | 0.980745 | 0.957393 | **0.968928** | 0.990146 | 0.992391 | 0.938863 |
| FCNN | 0.968045 | 0.977011 | 0.958647 | **0.967742** | 0.989988 | 0.991918 | 0.936256 |
| SVM | 0.968045 | 0.979461 | 0.956140 | **0.967660** | 0.988149 | 0.991014 | 0.936356 |
| LSTM | 0.961153 | 0.956576 | 0.966165 | **0.961347** | 0.986301 | 0.987585 | 0.922352 |
| NB | 0.960526 | 0.955390 | 0.966165 | **0.960748** | 0.989931 | 0.992488 | 0.921111 |
| KNN | 0.895363 | 0.846323 | 0.966165 | **0.902282** | 0.972934 | 0.974146 | 0.798776 |
| Lexicon | 0.875313 | 0.995041 | 0.754386 | **0.858161** | 0.875384 | 0.936210 | 0.773593 |

*Note*: The results are sorted by the preferred metric, highlighted in bold font.

relative performance of the lexicon classifier was higher for the imbalanced dataset. This is likely because the proposed keywords better fit the collected dataset of UN organizations, rather than the Kaggle dataset containing all kinds of tweets.

In the new setting, the ranking of the best-performing methods has changed. The number one is now CNN with $F_1 \approx 97\%$. This indicates that CNN might be an appropriate choice for classification with balanced datasets when the best possible performance needs to be squeezed out at a cost of higher complexity. In Table 4, CNN is closely followed by the RF and LR classifiers. It is however unclear whether the differences in the performance of these classifiers is statistically significant or not. To investigate this, we conduct pair-wise McNemar's tests [87] on every possible pair of classifiers under consideration. The corresponding methodology is described in the S1 Appendix, where Fig 8 in the S1 Appendix presents the results of the statistical tests. The results show that the performance of almost all the considered methods does not significantly differ from one another, with the exception of KNN and lexicon classifiers. That is, for scenarios where the dataset is balanced, it matters less which classification algorithm is chosen (with the exception of the KNN and lexicon classifiers).

We also sought to investigate the role of the imbalance further. To do this, we sweep the degree of balance in the dataset, adding new data from the Kaggle dataset to the training and test sets, keeping the same class proportions in both. We then compute the F1 scores for all the considered methods (see Fig 8). The dataset imbalance is captured by the balance ratio, which is defined as the ratio of positive to negative labels in the dataset. As mentioned before, for our original collected dataset, this ratio is only 0.09, hence it is heavily imbalanced. For the new completely balanced dataset, the balance ratio becomes 1.0. For each balance-ratio point in this experiment, we keep the balance between the classes the same for the training, validation and test sets.

Fig 8 shows that the performance of all classification algorithms improves with a higher balance ratio. The difference in performance diminishes as the balance ratio increases. This underpins the above observation that for nearly balanced datasets, the classification algorithms tend to perform similarly, with the exception of less capable KNN and lexicon classifiers. At the same time, the same algorithms—namely LR, RF and, to some extent, SVM and LSTM—perform best for almost all degrees of balance.

Taken together, we conclude that, for short text classification, it appears to make more sense to turn to traditional ML algorithms instead of computationally-demanding deep neural architectures. The benefits of the latter are marginal, while their training requires significantly

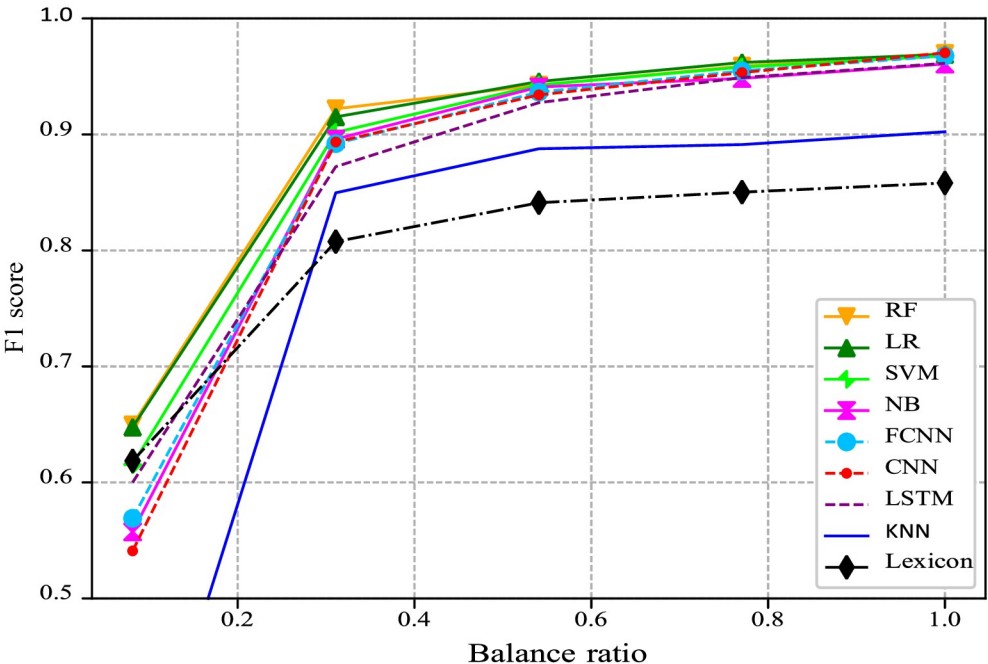

**Fig 8. F1 score vs. balance ratio for the classifiers under consideration.** Solid lines show the performance of traditional ML methods, dashed lines show that of DL-based approaches, while the dash-dotted line illustrates the performance of the unsupervised lexicon-based classifier.

more time. In contrast, DL methods might be more useful when datasets are balanced and of a large scale, and when extensive computational resources are available.

## Conclusions

In this article, we have compared the performance of various methods for the automated categorization of short texts in a small and imbalanced dataset. This classification task frequently occurs in social science research when researchers analyse short texts such as sentences, paragraphs, tweets, or article abstracts. Moreover, it is often the case that the concepts that are being examined are only present in a small proportion of the overall dataset. We have reviewed the most widely used text classifiers and compared their performance in this challenging setting. For this purpose, we have used a novel dataset on the communication about climate change on Twitter by eight international organizations operating in the UN system in different policy areas.

The article has made three main contributions to social science research engaging in automated text classification. First, it has provided a systematic overview of central text classification methods, including lexicons, traditional machine learning, and deep learning. In doing so, the article helps to reduce the entry barrier for researchers seeking to use such methods in their work. Computer science is a vast field and it is important to filter for the state-of-the-art methods that can be used to advance various fields in the social sciences.

Second, the performance assessment suggests that among the aforementioned classification methods, lexicons are clearly outperformed by machine-learning methods. However, it is important to take into account the role of class imbalance. For instance, the use of a convolutional neural network was found to be the best choice when there is a balanced dataset, but for

an imbalanced dataset, its performance is rather mediocre. We therefore recommend that researchers balance their datasets, for instance by complementing the dataset with more labeled data and by applying techniques of active learning [88]. Classifiers trained on balanced datasets are shown to perform much better.

Third, we have shown that simple traditional supervised classifiers, such as random forest and logistic regression, tend to perform as well as advanced neural networks in the aforementioned settings, while having much lower computational complexity. It therefore seems warranted to use deep learning only in those cases where there is a lot of high-quality data and extensive computational resources.

All told, future research should study the application of advanced methods, such as pretrained embeddings [89] and transformers [90], as well as novel hyperparameter tuning techniques [73], in combination with the methods discussed in this article. In the rapidly changing textual data landscape, the need for a systematic understanding of how the prominent methods for coding political texts perform is becoming ever more pressing.

## Supporting information

**S1 Appendix.**
(ZIP)

## Acknowledgments

The authors would like to thank to Hugo de Vos, Nicholas Olczak, and Maryna Povitkina for helpful comments. Earlier version of this paper was presented at the 2021 Annual Meeting of the American Political Science Association.

## Author Contributions

**Conceptualization:** Karina Shyrokykh, Max Girnyk.

**Data curation:** Karina Shyrokykh, Max Girnyk, Lisa Dellmuth.

**Formal analysis:** Karina Shyrokykh, Max Girnyk.

**Funding acquisition:** Lisa Dellmuth.

**Investigation:** Karina Shyrokykh, Lisa Dellmuth.

**Methodology:** Karina Shyrokykh, Max Girnyk.

**Project administration:** Lisa Dellmuth.

**Resources:** Lisa Dellmuth.

**Software:** Max Girnyk.

**Supervision:** Karina Shyrokykh, Lisa Dellmuth.

**Validation:** Max Girnyk.

**Visualization:** Max Girnyk.

**Writing – original draft:** Karina Shyrokykh, Max Girnyk, Lisa Dellmuth.

**Writing – review & editing:** Lisa Dellmuth.

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
