## [Decision Letter · Decision Letter 0]

5 Feb 2023

PONE-D-22-35683Short Text Classification with Machine Learning in the Social Sciences: The Case of Climate Change on TwitterPLOS ONE

Dear Dr. Shyrokykh,

Thank you for submitting your manuscript to PLOS ONE. After careful consideration, we feel that it has merit but does not fully meet PLOS ONE’s publication criteria as it currently stands. Therefore, we invite you to submit a revised version of the manuscript that addresses the points raised during the review process. Dear authors, I found your manuscript very interesting, however there are some concerns that should be addressed before publication. Please revise your manuscript carefully according to all reviewers' comments. On top of those comments, I would like to add the following: - Please conduct statistical tests, as this is the only way to prove that results generated by one method are significantly better than the ones obtained by  other approaches included in comparative analysis.-  The GITHUB link with dataset and source code that you provided is broken. Please, make sure that you share all relevant details, including the dataset and the source code, so that other researchers' could try to even further improve achieved results.

We look forward to receiving your revised manuscript.

Kind regards,

Nebojsa Bacanin

Academic Editor

PLOS ONE

Journal Requirements:

"This work was supported by the project: “Glocalizing Climate Governance: The role of Integrated Governance for a Just and Legitimate Adaptation to Climate Risks (GlocalClim)” funded by Formas under grant number 2018-01705. Svenska Forskningsrådet Formas, Grant/Award Number: 2018-01705."

4. We note that Supporting Figure 1 in your submission contain copyrighted images. All PLOS content is published under the Creative Commons Attribution License (CC BY 4.0), which means that the manuscript, images, and Supporting Information files will be freely available online, and any third party is permitted to access, download, copy, distribute, and use these materials in any way, even commercially, with proper attribution. For more information, see our copyright guidelines: http://journals.plos.org/plosone/s/licenses-and-copyright.

(1) You may seek permission from the original copyright holder of Supporting Figure 1 to publish the content specifically under the CC BY 4.0 license. 

**Additional Editor Comments:**

Dear Authors,

Please revise your paper carefully according to reviewers' comments.

All the best,

AE

Reviewers' comments:

Reviewer's Responses to Questions

**Comments to the Author**

1. Is the manuscript technically sound, and do the data support the conclusions?

Reviewer #1: Yes

Reviewer #2: Yes

Reviewer #3: Yes

2. Has the statistical analysis been performed appropriately and rigorously? 

Reviewer #1: Yes

Reviewer #2: Yes

Reviewer #3: No

3. Have the authors made all data underlying the findings in their manuscript fully available?

Reviewer #1: Yes

Reviewer #2: Yes

Reviewer #3: No

4. Is the manuscript presented in an intelligible fashion and written in standard English?

Reviewer #1: Yes

Reviewer #2: Yes

Reviewer #3: Yes

5. Review Comments to the Author

Reviewer #1: 1. Avoid using we/our throughout the manuscript.

2. Literature review should be expanded with recent machine learning models tuned by metaheuristics optimizer, that were applied to the text classification problem. Include the following references:

https://www.mdpi.com/2227-7390/9/16/1929

https://link.springer.com/chapter/10.1007/978-981-19-3035-5_56

https://www.tandfonline.com/doi/full/10.1080/08839514.2021.2004345

https://www.mdpi.com/2227-7390/10/22/4173

https://www.inderscienceonline.com/doi/abs/10.1504/IJCAT.2022.123237

3. Line 39, unsupervised, line 44, supervised, and line 47, deep learning - is it necessary to mark them in italic font?

4. Add the structure of the paper at the end of the introduction section.

5. Make sure that each parameter in every equation has been explained in the text.

6. Line 155 - again, is it necessary to use italic font? Also consider is it required to use it in other places throughout the text.

7. Figure 1 is not adding anything valuable to the manuscript, and should be removed.

8. In my opinion, tables with the results should not be sorted by the preferred metric. Rather, highlight the best result in each category in bold

For example, in Table 2, LR F1 score, KNN precision, SVM recall etc.

9. Discuss the limitations of each observed classifier.

Reviewer #2: The article deals with text classification using ML models.

I have following suggestions to the authors:

1. The main contribution of the paper is comparison between various ML models. The authors are reviewing different automated text classification methods. They use different optimizers for hyperparameter tuning. Having that in mind, authors should mention (or even consider using in the future work) recent articles which use ML models and metaheuristics (for parameter tuning and feature selection) in language processing, for example:

Bacanin, N.; Zivkovic, M.; Stoean, C.; Antonijevic, M.; Janicijevic, S.; Sarac, M.; Strumberger, I. Application of Natural Language Processing and Machine Learning Boosted with Swarm Intelligence for Spam Email Filtering. Mathematics 2022, 10, 4173. https://doi.org/10.3390/math10224173

Nandanwar, A.K.; Choudhary, J. Semantic Features with Contextual Knowledge-Based Web Page Categorization Using the GloVe Model and Stacked BiLSTM. Symmetry (Basel). 2021, 13, 1772

El-Alami, F. zahra; Ouatik El Alaoui, S.; En Nahnahi, N. Contextual Semantic Embeddings Based on Fine-Tuned AraBERT Model for Arabic Text Multi-Class Categorization. J. King Saud Univ. - Comput. Inf. Sci. 2022, 34, 8422–8428

Reviewer #3: The paper puts forward a data set with tweets from 8 UN organizations. The statements are manually labelled as being or not related to climate change. The data set itself represents an important advancement not only for replicating the current experiment, but even for being used as benchmark in the future. However, the link provided for the data set is not valid. The authors are advised to use a platform like figshare.com for providing the data (I underline that the current web link is not even valid).

The experiments are nicely described. Particularly, the experiment with the controlled balance ratio was very interesting.

Please clarify whether the same balance between the classes that exists in the entire data set is also kept in all data sets, training, validation and test.

Please also provide more information on the parameters that were used within the parameter tuning for each model in turn. The conclusion regarding the better and worse models is quite strong and it may be the case that the parameter tuning might be more advantageous for some models than for others.

Some statistical tests could highlight whether the differences in results are significant or not, be that there are enough results for such tests.

6. PLOS authors have the option to publish the peer review history of their article (what does this mean?). If published, this will include your full peer review and any attached files.

Reviewer #1: No

Reviewer #2: No

Reviewer #3: No

---

## [Author Response · Author response to Decision Letter 0]

11 Aug 2023

See the attached rebuttal file with point-to-point responses

---

## [Decision Letter · Decision Letter 1]

16 Aug 2023

Short Text Classification with Machine Learning in the Social Sciences: The Case of Climate Change on Twitter

PONE-D-22-35683R1

Dear Dr. Shyrokykh,

We’re pleased to inform you that your manuscript has been judged scientifically suitable for publication and will be formally accepted for publication once it meets all outstanding technical requirements.

Kind regards,

Nebojsa Bacanin

Academic Editor

PLOS ONE

Additional Editor Comments (optional):

Dear Authors,

I think that the manuscript can now be accepted.

Warmest,

AE

Reviewers' comments:

Reviewer's Responses to Questions

**Comments to the Author**

1. If the authors have adequately addressed your comments raised in a previous round of review and you feel that this manuscript is now acceptable for publication, you may indicate that here to bypass the “Comments to the Author” section, enter your conflict of interest statement in the “Confidential to Editor” section, and submit your "Accept" recommendation.

Reviewer #1: All comments have been addressed

Reviewer #2: All comments have been addressed

2. Is the manuscript technically sound, and do the data support the conclusions?

Reviewer #1: (No Response)

Reviewer #2: Yes

3. Has the statistical analysis been performed appropriately and rigorously? 

Reviewer #1: (No Response)

Reviewer #2: Yes

4. Have the authors made all data underlying the findings in their manuscript fully available?

Reviewer #1: (No Response)

Reviewer #2: Yes

5. Is the manuscript presented in an intelligible fashion and written in standard English?

Reviewer #1: (No Response)

Reviewer #2: Yes

6. Review Comments to the Author

Reviewer #1: (No Response)

Reviewer #2: All comments have been addressed properly.

I recommend the manuscript for publication.

7. PLOS authors have the option to publish the peer review history of their article (what does this mean?). If published, this will include your full peer review and any attached files.

Reviewer #1: No

Reviewer #2: No

---

## [Editor Report · Acceptance letter]

21 Sep 2023

PONE-D-22-35683R1 

Short text classification with machine learning in the social sciences: The case of climate change on Twitter 

Dear Dr. Shyrokykh:

I'm pleased to inform you that your manuscript has been deemed suitable for publication in PLOS ONE. Congratulations! Your manuscript is now with our production department. 

Kind regards, 

on behalf of

Dr. Nebojsa Bacanin 

Academic Editor

PLOS ONE